# Methimazole-Induced ANCA Vasculitis: A Case Report

**DOI:** 10.3390/diagnostics11091580

**Published:** 2021-08-31

**Authors:** Precil Diego Miranda de Menezes Neves, Lucas Braga Mota, Cristiane Bitencourt Dias, Luis Yu, Viktoria Woronik, Lívia Barreira Cavalcante, Denise Maria Avancini Costa Malheiros, Lectícia Barbosa Jorge

**Affiliations:** 1Nephrology Division, School of Medicine, University of São Paulo, São Paulo 01246-903, SP, Brazil; lucasbraga3@gmail.com (L.B.M.); cristianebitencourt@uol.com.br (C.B.D.); luisyu@usp.br (L.Y.); viktoriaw@usp.br (V.W.); lecticiajorge@gmail.com (L.B.J.); 2Nephrology and Dialysis Service, Oswaldo Cruz German Hospital, São Paulo 01323-020, SP, Brazil; 3Pathology Division, School of Medicine, University of São Paulo, São Paulo 01246-903, SP, Brazil; liviabarreira@gmail.com (L.B.C.); denise.mac.malheiros@gmail.com (D.M.A.C.M.)

**Keywords:** ANCA, ANCA associated vasculitis, crescentic glomerulonephritis, vasculitis, alveolar hemorrhage, methimazole, antithyroid drugs, immunosuppression therapy

## Abstract

Rapidly progressive glomerulonephritis (RPGN) is a syndrome which presents rapid loss of renal function. Vasculitis represents one of the major causes, often related to anti-neutrophil cytoplasmic antibodies (ANCA). Herein, we report a case of methimazole-induced ANCA-associated vasculitis. A 35-year-old woman complained of weight loss and fatigue for 2 weeks and attended the emergency room with alveolar hemorrhage. She had been diagnosed with Graves’ disease and had been taking methimazole in the past 6 months. Her physical examination showed pulmonary wheezing, hypertension and signs of respiratory failure. Laboratory tests revealed urea 72 mg/dL, creatinine 2.65 mg/dL (eGFR CKD-EPI: 20 mL/min/1.73 m^2^), urine analysis with >100 red blood cells per high-power field, 24 h-proteinuria: 1.3 g, hemoglobin 6.6 g/dL, white-cell count 7700/mm^3^, platelets 238,000/mm^3^, complement within the normal range, negative viral serological tests and ANCA positive 1:80 myeloperoxidase pattern. Chest tomography showed bilateral and diffuse ground-glass opacities, and bronchial washing confirming alveolar hemorrhage. A renal biopsy using light microscopy identified 27 glomeruli (11 with cellular crescentic lesions), focal disruption in glomerular basement membrane and fibrinoid necrosis areas, tubulitis and mild interstitial fibrosis. Immunofluorescence microscopy showed IgG +2/+3, C3 +3/+3 and Fibrinogen +3/+3 in fibrinoid necrosis sites. She was subsequently diagnosed with crescentic pauci-immune glomerulonephritis, mixed class, in the setting of a methimazole-induced ANCA vasculitis. The patient was treated with methimazole withdrawal and immunosuppressed with steroids and cyclophosphamide. Four years after the initial diagnosis, she is currently being treated with azathioprine, and her exams show creatinine 1.30 mg/dL (eGFR CKD-EPI: 52 mL/min/1.73 m^2^) and negative p-ANCA.

## 1. Introduction

Vasculitis is a heterogeneous group of autoimmune diseases which can be classified based on the caliber of the vessels involved as large, medium or small size, and affects patients at different ages. The existence of anti-neutrophil cytoplasmic antibodies (ANCA) in some of these pathologies leads to ANCA associated vasculitis (AAV) [1].

AAV mainly affects older patients, with an onset peak between 60 and 70 years old. AAV has an autoimmune and inflammatory background and can be triggered by some viruses, bacteria or drugs. The most frequently prescribed medications associated with ANCA and AAV are hydralazine, allopurinol and antithyroid drugs (propylthiouracil and methimazole) [2,3]. Renal involvement is frequent in AAV, presenting rapidly progressive glomerulonephritis (RPGN) and comprising granulomatosis with polyangiitis (GPA), microscopic polyangiitis (MPA) and eosinophilic GPA (EGPA) [2,3,4,5].

The histopathological classification of crescentic glomerulonephritis includes four classes: focal, crescentic, mixed and sclerotic; each one of them presents a direct correlation to renal survival and the necessity of renal replacement therapy [6,7]. Treatment includes immunosuppression and drug withdrawal in cases when a suspect drug is in use [8].

Herein, we describe a case of a young female patient presenting pulmonary kidney syndrome, secondary to AAV induced by methimazole, for whom early diagnosis and treatment enabled a favorable evolution and recovery of renal function.

## 2. Case Report

A 33-year-old female attended the emergency room 4 years ago complaining of shortness of breath for two days and pulmonary hemorrhage. She described intermittent coughing in the past three months, weight loss of 5 kg in the last two weeks, asthenia and anorexia. She denied any history of respiratory diseases or contact with tuberculosis. The patient had been diagnosed with Graves’ disease six months prior and was on methimazole 10 mg per day and propranolol 40 mg per day.

She appeared unwell upon examination, presenting respiratory sounds with bilateral wheezing, increased thyroid gland, a palpable goiter, no nodules and a physical examination with no otherwise relevant findings. The respiratory rate was 32 breaths per minute, oxygen saturation 83% while she was breathing ambient air, and blood pressure 160/90 mm Hg. She was promptly submitted to endotracheal intubation. Laboratory tests revealed urea 72 mg/dL, and creatinine 2.65 mg/dL (eGFR CKD-EPI: 20 mL/min/1.73 m^2^). Urine analysis revealed more than 100 red blood cells per high-power field, 24 h-proteinuria: 1.3 g, no serum electrolyte or acid-base disorder, albumin was 2.1 g/dL, cholesterol 172 mg/dL, triglycerides 116 mg/dL, hemoglobin 6.6 g/dL, white-cell count 7700 per microliter, platelets 238,000 per microliter, C-reactive protein 96 mg/dL, erythrocyte sedimentation rate (ESR) 89 mm/h, C3 was 103 mg/dL and C4 was 18 mg/dL (complement within the normal range). Serological tests for hepatitis B, C, HIV and rheumatoid factor were negative. There was no evidence of monoclonal proteins. Posterior exams revealed positive p-ANCA 1:80, and antinuclear antibodies (ANA) 1:80. She had performed exams 2 months prior to admission presenting creatinine of 0.69 mg/dL (eGFR CKD-EPI: 114 mL/min/1.73 m^2^), and hemoglobin 12.4 g/dL, highlighting a disease with acute and rapid loss of renal function and severe anemia (Table 1).

Ultrasound revealed normal-sized kidneys with preserved echogenicity. A chest CT scan evidenced bilateral and diffuse ground-glass opacities (Figure 1), thickening of intralobular septa and luminal filling in the bronchial walls, compatible with alveolar hemorrhage. She was prescribed antibiotics (ceftriaxone and clarithromicin) due to possible bacterial infection. Serum immunodiffusion for fungi was negative (*Paracoccidiodes brasiliensis*, *Aspergillus fumigatus* and *Histoplasma capsulatum*). Bronchial washing results were negative for viruses, bacteria, mycobacteria and fungi, revealing the presence of bloody fluid in the respiratory tract.

A diagnosis of pulmonary-renal syndrome was established since bronchoscopic findings showed no evidence of pulmonary infection. We suspended antibiotics, initiated intravenous (IV) metilprednisolone 1 g per day for 3 days and proceeded to perform a renal biopsy. A total of 27 glomeruli were identified by light microscopy (5 globally sclerotic and 11 with cellular crescentic lesions), evidence of focal rupture in glomerular basement membrane with fibrinoid necrosis areas (Figure 2). Dilated tubules with regenerative epithelium, focal atrophy and lymphocytes attacking the basement membrane. Focal interstitial fibrosis with lymphocytic and plasmocytic infiltration, and focal edema. Arched artery sample with intimal fibrosis and preserved arterioles. Immunofluorescence microscopy detected IgG (+2/+3), C3 (+3/+3) and Fibrinogen (+3/+3), with mesangial deposition and a granular pattern, as well as segmental and focal distribution in fibrinoid necrosis sites (Figure 2). Biopsy findings were compatible with crescentic pauci-immune glomerulonephritis, mixed class, in the setting of a methimazole-induced ANCA vasculitis.

The patient was treated with methimazole withdrawal, and the patient was put on cyclophosphamide treatment as complementary therapy to IV metilprednisolone. Due to rapid pulmonary improvement after those treatments, we decided not to perform plasmapheresis and maintain cyclophosphamide monthly for 6 months. The patient showed improvement in renal function and proteinuria after induction treatment, with resolution of anemia and hematuria and negative ANCA.

The patient then became pregnant two years after the diagnosis. She tested positive for ANCA (1:80) again during pregnancy, but without worsening renal function or proteinuria (Table 1). Thus, the use of azathioprine, oral prednisone and aspirin were maintained. The patient gave birth at 37 weeks, without any complications.

Four years after the diagnosis, the patient currently performs periodic visits to an outpatient clinic, she is on maintenance therapy with azathioprine and in complete renal and pulmonary remission. Recent laboratory exams revealed that her creatinine was 1.30 mg/dL (eGFR CKD-EPI: 52 mL/min/1.73 m^2^), protein to creatinine ratio 0.2 g/g, hemoglobin 14.2 g/dL, negative p-ANCA values and normal complement levels. Thyroid disease is controlled by endocrinologist physicians at the same hospital after successful radioactive iodine therapy.

## 3. Discussion and Conclusions

Systemic vasculitis can be classified based on the caliber of vessel involvement as large, medium or small size [1]. A group of this vasculitis encompasses the anti-neutrophil cytoplasmatic antibody (ANCA) associated vasculitis (AAV), manifesting antibodies directed against myeloperoxidase (MPO) and proteinase 3 (PR3). ANCA is often detected using two techniques: direct immunofluorescence and ELISA. The indirect immunofluorescence (IF) reaction is performed by exposing the patient’s serum with human neutrophils fixed in ethanol. In this method, we can observe the C-ANCA pattern with positivity in all the cytoplasm, in most cases reflecting the presence of anti-PR3 antibodies. The other pattern that can be observed in IF is P-ANCA, which reflects the positivity of the perinuclear exam, which is most commonly associated with the presence of anti-MPO antibodies. The other way of detecting ANCA is by ELISA through detecting specific antibodies to MPO or PR3. The first technique tends to be more sensitive and the second more specific. In the case presented herein, ANCA was detected using the indirect IF technique, since we did not have the ELISA detection test available in our institution at the time of diagnosis, instead detecting the P-ANCA pattern, which more frequently correlates with anti-MPO antibodies [9,10,11]. The AAV spectrum which affects small size vessels with renal involvement comprehends granulomatosis with polyangiitis (GPA), microscopic polyangiitis (MPA) and eosinophilic GPA (EGPA) [2,3].

AAV generally affects older patients with an onset peak between 60 and 70 years old. The incidence varies in 10 up to 25 cases per million habitants in Europe and United States, having a higher prevalence in European and Asian ancestry, and in the male gender. AAV has an autoimmune and inflammatory background, and can be triggered by some viruses, bacteria or drugs. The most frequently prescribed medications associated with ANCA and AAV are hydralazine, allopurinol and antithyroid drugs (propylthiouracil and methimazole) [2,3,4].

The pathogenesis of AAV is complex. It is postulated that genetic factors predispose the patient to produce antibodies leading to activation of neutrophils and monocytes, causing endothelial damage, cytokine production and activation of the alternative complement pathway. A second hypothesis theorizes that some alterations are necessary to cause the interaction between the circulating antibodies and the antigen sites. This second event can occur as a consequence of environmental triggers such as infections, drugs or other undetermined stimuli [2,3,4].

Renal involvement is frequent in AAV. Patients present rapidly progressive glomerulonephritis (RPGN) with acute kidney injury, hematuria and proteinuria in an accelerated manner. Kidney biopsy findings encompass cellular, fibrocellular and fibrous crescents, rupture of the glomerular basement membrane, fibrinoid necrosis, glomerulonephritis, interstitial nephritis and glomerular sclerosis. Immunofluorescence (IM) classically presents a pauci-immune pattern, but immune complexes and immunoglobulins can be found in some cases. Electron microscopy adds exiguous findings to clinical practice [3,5,6,7].

The histopathologic classification includes four classes: focal, crescentic, mixed and sclerotic. These classes present a direct correlation to renal survival and the necessity of renal replacement therapy. An investigation regarding renal outcomes describes that the renal survival percentages at 1 year were 93% for biopsies classified as focal, 84% for biopsies classified as crescentic, 69% for mixed form and 50% for patients with sclerotic biopsy identification [7]. Similarly, a series of 58 pauci-immune glomerulonephritis cases followed by Ramalho et al. (unpublished data), in which the patients’ mean age was 45 years, 37% male and with ANCA positivity in about 37%, identified the worst outcomes as being related to the sclerotic and the crescentic types. Large cohorts in the literature have shown markedly poor prognosis related to sclerotic and crescentic subtypes. It is commonly reported that focal subtype is usually associated with a better prognosis compared to the other forms [12,13].

AAV treatment consists of an induction phase combining steroids with cyclophosphamide or rituximab. Plasmapheresis should also be considered in patients manifesting pulmonary hemorrhage. Maintenance therapy should be performed with low dose steroids associated with azathioprine, rituximab or mycophenolate for at least 24 months [14,15,16]. History of drug use must be kept in mind for AAV patients. In these cases, treatment includes drug withdrawal and appropriate immunosuppression due to the severity of the clinical manifestations and potential renal function recovery. Cutaneous manifestations are the most common presentation of drug induced vasculitis and other symptoms include myalgia, arthralgia, malaise, fatigue and fever. Antibody titers tend to present negative in about six months after drug discontinuation, as evidenced in the case presented herein [3,16].

Commonly associated medications are hydralazine, allopurinol, levamisole, penicillamine, sulfasalazine and antithyroid drugs (ATD) [17]. The likelihood of ANCA development increases with the time of antithyroid drug exposure, as well as the possibility of AAV. The presence of ANCA occurs in about 10% in patients taking ATD in the first 8 months of drug use and increases to 27% with chronic use. The main antibody category in these cases is directed against myeloperoxidase (anti-MPO) [18]. The prevalence of ANCA was 1.8 times higher in a series of patients using ATD than in the general population, not necessarily resulting in AAV.

The specific pathogenesis of AAV associated with ATD is poorly understood, but it is postulated that the drug and its metabolites could accumulate in neutrophils, binding to myeloperoxidase, altering its conformation and inducing ANCA production. The ATD oxidization in the presence of activated neutrophils creates reactive specimens which stimulate ANCA development. Differently, propylthiouracil could decrease the degradation of neutrophil traps, resulting in more antigen exposure and autoimmunity phenomena [2,3,4]. Drug withdrawal is a cornerstone of AAV treatment associated with immunosuppressive therapy, similar to other AAV cases not related to drug exposure [19].

The patient became pregnant during maintenance therapy and delivered a healthy baby without complications. A rise in pregnancy incidence in patients with primary vasculitis has been described in recent years due to intensive disease control and better outcomes, but little is known about the course of the disease at the time that some patients could present worsening or improvement in clinical features and in kidney function during pregnancy. The results related to the fetus are also uncertain, with case reports showing diverse outcomes [20,21,22].

One study reviewed the literature and found 97 pregnancies in patients with small vessel vasculitis (GPA, EGPA and MPA). Common complications included 25 premature deliveries (25%), nine spontaneous abortions (9%) and cases of pre-eclampsia, hemorrhage and placental hematoma, and other fetal complications. Vasculitis diagnosis was frequently made before and after pregnancy, and only rarely during the pregnancy months. Childbearing could be associated with disease flares or disease remission in varying scenarios in the literature. The disease course in patients with vasculitis is not clear due to low prevalence of these pathologies and an association with gender and age, and therefore more research and studies are needed to evaluate the pregnancy prognosis and other maternal and fetal consequences [22].

Vasculitis is a rare disease in clinical practice, but early diagnosis and proper management are decisive to provide better care and to improve clinical outcomes. Medications are a possible trigger associated with AAV and should be promptly discontinued in these cases. Women in childbearing age can be affected by small vessel vasculitis and pregnancy may have various courses, with possible complications to the patient and to the fetus.

## Figures and Tables

**Figure 1 diagnostics-11-01580-f001:**
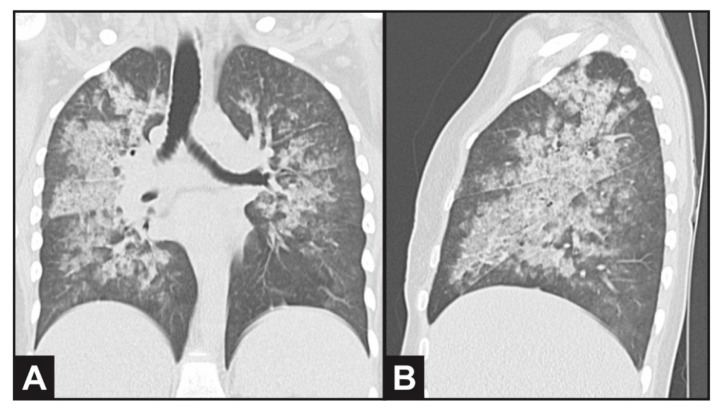
Computed thorax tomography evidencing bilateral and diffuse ground-glass opacities in (**A**) coronal and (**B**) sagittal plane view, compatible with alveolar *hemorrhage*.

**Figure 2 diagnostics-11-01580-f002:**
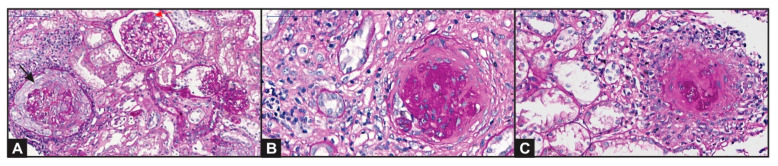
Kidney biopsy showing: (**A**) a circumferential cellular crescent producing compressive effect on glomerular tuft (black arrow). Some remaining glomeruli maintain their normal structure or segmental sclerotic lesions (red arrow)—Periodic Acid-Shiff (PAS); (**B**) a sclerotic crescent surrounding a sclerotic glomerulus (PAS); (**C**) mononuclear inflammatory cells around a glomerulus with capsule rupture, revealing a “granulomatous-like” or “sunburst” aspect (PAS).

**Table 1 diagnostics-11-01580-t001:** Laboratory tests along clinical follow-up.

Laboratory Test	2 Months Prior to Renal Biopsy	At Renal Biopsy	Post-Treatment (6 Months after Renal Biopsy)	Pregnancy (2 Years after Renal Biopsy)	Currently (4 Years after Renal Biopsy)
Creatinine (mg/dL)	0.69	2.65	1.42	1.09	1.3
eGFR CKD-EPI (mL/min)	114	20	46.5	65.4	52.6
Urea (mg/dL)	23	72	36	46	40
Hemoglobin (g/dL)	12.4	7.8	13.1	12.2	14.2
Proteinuria (g/day)	N/A	1.3	0.26	0.36	0.2
Hematuria	N/A	>100 RBC/HPF	Negative	Negative	Negative
ANCA (titles)	N/A	1:80	Negative	1:80	Negative

ANCA: anti-neutrophil cytoplasmic antibodies; HPF: high power field; N/A: not available; RBC: red blood cells.

## Data Availability

Data and material are available on request.

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
