# Peer review of "Methimazole-Induced ANCA Vasculitis: A Case Report"

_diagnostics, 2021, doi:10.3390/diagnostics11091580_

Round 1
Reviewer 1 Report
I had the oportunity to review the manuscript by Precil Diego Miranda de Menezes Neves, et al entitled Methimazole-induced ANCA vasculitis: a case report whisn is intended to be published in Diagnostics.
The manuscript is well written and the case well presented, however it does not sound as original. The presence of ANCA occurs in about 10% in patients taking ATD in the first months of drug use and increases to 27% with chronic use.
The only interesting but weak data is a pleiomorphic positive immunofluorescence which appears as C3 glomerulopathy. But few novelty
Author Response
Thank you for the kind comments about the manuscript. Although the case is not unprecedented, the description of the challenges to reach the diagnosis as well as the unusual pattern of immunofluorescence and the recovery of the patient's renal function make it interesting and of great clinical applicability. Regarding immunofluorescence, in a minority of cases, even in the setting of an ANCA-associated vasculitis, the presence of positivity for IgG and C3 may be observed in the renal biopsy, especially in the glomerular necrosis areas, as described in the discussion sesction.Reviewer 2 Report
The paper by Mirande de Menezes Neves et al is a clearly written case report on the well known complication of methamizol ANCA vasculitis. Their report is not novel but adds well to the existing literature and the kidney biopsy photographs are nice. I have some small comments:
Comments
I think the MPO-ANCA ELISA was not determined and instead an immunofluorescence test was done which showed a p-ANCA. The authors should explain the overlap of p-ANCA with MPO-ANCA before discussing the role of MPO-ANCA in AAV in the discussion.
Why did this patient not receive rituximab? Was that a financial reason?
line 40: AAR should be AAV
line 200: the terms Wegener's and Churg-Strauss are not used anymore, this should be called GPA and EGPA. This had to do with the fascistic past of Friedrich Wegener during the second world war and the general process of getting rid of using people's names to describe diseases.
Author Response
1. I think the MPO-ANCA ELISA was not determined and instead an immunofluorescence test was done which showed a p-ANCA. The authors should explain the overlap of p-ANCA with MPO-ANCA before discussing the role of MPO-ANCA in AAV in the discussion.
Response: Thank you for your comment. In our study, ANCA detection was performed by indirect immunofluorescence since the analysis by the ELISA was not available in our service at that time. We discussed about the overlap of p-ANCA with MPO-ANCA in the discussion section as requested (Page 4, lines 129-141, highlighted in yellow)
2. Why did this patient not receive rituximab? Was that a financial reason?
Response: Thank you for the raised point. As previously stated by the reviewer, the patient was not treated with Rituximab due to the unavailability of routine use in our service due to financial reasons.
3. line 40: AAR should be AAV
line 200: the terms Wegener's and Churg-Strauss are not used anymore, this should be called GPA and EGPA. This had to do with the fascistic past of Friedrich Wegener during the second world war and the general process of getting rid of using people's names to describe diseases.
Response: Thank you for your relevant comment. We made the correction of the terms in the manuscript as suggested (Page 1, line 40 and Page 5, line 213, both highlighted in yellow)
Round 2
Reviewer 1 Report
No more comments. The study is ok but lacks novelty